# Study on the Relationship between Economic Growth of Animal Husbandry and Carbon Emission Based on Logarithmic Average Index Method and Decoupling Model: A Case Study of Heilongjiang Province

Tao He [1,2], Xiuwei Lin [3], Yongli Qu [1,2] and Chunbo Wei [1,2,*]

[1] Key Laboratory of Low-Carbon Green Agriculture in Northeastern China, Ministry of Agriculture and Rural Affairs, Beijing 100125, China; ht960704@163.com (T.H.); ylqu007@126.com (Y.Q.)

[2] Department of Animal Science, College of Animal Science and Veterinary Medicine, Heilongjiang Bayi Agricultural University, Daqing 163316, China

[3] Branch of Animal Husbandry and Veterinary of Heilongjiang Academy of Agricultural Sciences, Qiqihaer 163005, China; lxw960521@163.com

[*] Correspondence: weichunbo@byau.edu.cn

**Abstract:** With the establishment of the action plan for the goal of "carbon peaking and carbon neutrality", how to achieve high-quality agricultural development, help implement the construction of the green Longjiang River, reduce agricultural carbon emissions, and increase the level of agricultural carbon sink is a key problem that must be solved for Heilongjiang Province to achieve the goal of "double carbon". This article uses the Life Cycle Assessment (LCA) method to estimate the carbon emissions of animal husbandry in Heilongjiang Province and 13 cities from 2000 to 2020. By constructing the Tapio decoupling model, Kaya identity, and the LMDI model, the relationship between animal husbandry economy and carbon emissions in Heilongjiang Province and the driving factors affecting animal husbandry carbon emissions are explored. The results indicate that: (1) From 2000 to 2020, the carbon emissions of animal husbandry in Heilongjiang Province showed an overall slightly upward trend. From the perspective of various emission links, the highest carbon emissions are from the gastrointestinal fermentation environment (42.49%), with beef cattle, cows, and live pigs being the main livestock and poultry in Heilongjiang Province with carbon emissions. (2) The Tapio decoupling model results indicated that from 2000 to 2020, the relationship between the economic development of animal husbandry in Heilongjiang Province and carbon emissions was mainly characterized by weak decoupling. (3) The main driving force behind the continuous increase in carbon emissions from animal husbandry in Heilongjiang Province is the changing factors of agricultural population returns and changes in the production structure of animal husbandry; The driving factors that suppress the increase in carbon emissions from animal husbandry in Heilongjiang Province are changes in animal husbandry production efficiency, population and urban development levels, and population mobility factors. Finally, based on the decoupling effect status and driving factors of decomposition between Heilongjiang Province and its various cities, it is recommended to implement countermeasures and suggestions for the transformation of animal husbandry in the province towards green and low carbon at the macro level. This can be achieved through the adoption of sustainable and eco-friendly practices such as the use of renewable energy sources and the reduction of greenhouse gas emissions. Additionally, promoting research and development in sustainable agriculture and animal husbandry can also contribute to the transformation towards a more environmentally friendly industry.

**Keywords:** carbon emissions; life cycle assessment; tapio decoupling model; kaya identity; LMDI model

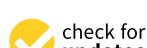



## 1. Introduction

In recent years, the number of natural disasters and extreme weather events around the world has been increasing year by year, reflecting a series of impacts caused by the

continuous changes in the climate and environment [1]. In response to the issue of climate change caused by carbon emissions, countries around the world have strengthened the governance of various aspects such as the environment, introduced corresponding policies and guidelines, and achieved some phased results [2]. Global carbon emissions have been increasing at an annual rate of over 3% since 2000, but the growth rate has slowed down since 2010. From 2014 to 2016, carbon emissions remained relatively balanced and only slightly increased, showing an upward trend from 2017 to 2018, and a slight decrease in 2019. Nevertheless, the current global carbon emissions situation remains severe [3].

Based on Organization of Economic Cooperation and Development (OECD) data, China has the highest carbon emissions globally, increasing from 5961.80 million tons in 2006 to 9876.50 million tons in 2019, with an average annual growth rate of 4% [4]. China, as the world's largest carbon emitter, faces a significant challenge in achieving decoupling of carbon emissions from its economy in the face of climate change. During the United Nations General Assembly on 22 September 2020, China made a commitment to implement robust policies and measures in order to achieve a carbon peak by 2030 and carbon neutrality by 2060. This bold move highlights China's unwavering dedication to promoting eco-friendly and low-carbon development. In recent years, there has been a growing focus on the academic community on the issue of carbon emissions. Using data released by the state, researchers have analyzed and explored various perspectives on the trend of China's future carbon peak. According to some scholars, China has demonstrated the capability to achieve the objective of 'carbon peak, carbon neutrality' as per the plan. However, there are still some scholars who view this goal as a challenging feat. For example, the research of Yang P and others show that according to China's '1 + N' dual carbon policy system, Hebei, Ningxia, and other regions need to clarify the carbon peak year as soon as possible to avoid leaving insufficient time to achieve carbon neutrality. At the same time, they also proposed that there are only 30 years from the carbon peak of carbon neutrality, so carbon neutrality action plans should be formulated as soon as possible [5]. In a recent study, Yang M et al. suggested that China's goal of achieving carbon neutrality by 2026 can be met by ensuring that the renewable energy ratio exceeds 80%, the urbanization rate reaches 85%, and energy consumption is limited to 6.5 billion tons [6]. Guo and XM's research indicates that the model results will follow an inverted 'U' shape trend over the next 20 years, with the peak being reached in 2036 [7].

Methane ($CH_4$) and Nitrous Oxide ($N_2O$) contribute to almost 90% of the total greenhouse gas emissions, with animal husbandry being a significant contributor. Animal husbandry has been identified as a significant contributor to greenhouse gas emissions [8,9]. As a result, there is a growing trend towards transforming the industry to become more environmentally friendly and low carbon. This shift towards a low-carbon animal husbandry approach, which prioritizes low consumption, low emissions, high efficiency, and ecological sustainability, has gained widespread support among government decision-makers in various countries [10].

Heilongjiang Province, situated in the northeast of China, boasts a vast territory and fertile soil. It is a crucial commodity grain base in China and an acclaimed province for the dairy industry. The province possesses abundant resources and unique geographical advantages, making it an ideal location for the development of animal husbandry [11].

In China, the development and trends of animal husbandry vary between provinces, leading to differences in livestock and poultry structure, quantity, and population mobility. To investigate the impact of animal husbandry structure on carbon emissions, Shi, R analyzed data from 30 provincial-level administrative regions in China between 2000 and 2018. The study utilized the Thiel index, density analysis, and convergence analysis to quantify the effects. They found that the carbon emissions of animal husbandry were the highest in agricultural areas (showing a downward trend), followed by agricultural and pastoral areas (showing a downward trend), and lowest in pastoral areas (showing an upward trend) [12]. Li and M utilized the ecological footprint method to comprehensively evaluate carbon emissions and carbon sequestration. They calculated the carbon

footprint of agriculture and animal husbandry in the Qinghai Tibet Plateau and analyzed its spatiotemporal distribution characteristics. Based on their findings, they proposed plans to reduce carbon emissions and suggested methods to improve carbon sinks [13]. Hao conducted a study on the carbon emissions generated by China's animal husbandry industry. The study utilized a bidirectional fixed effect model that controlled for time and space variables. The aim was to examine the impact of various factors on the supply side of animal husbandry on the dependent variable. The study findings indicate that the carbon emissions resulting from animal husbandry are significantly impacted by the land structure, aquaculture structure, technological level, and scale level. However, the impact of the human capital level and mechanization level is comparatively less significant as compared to the technological and scale structures [14]. In his proposal, Cai suggested that in order to achieve China's carbon intensity reduction target for 2030, it should be allocated to each province. He also explored the vertical linkage relationship between China and its 31 provinces' livestock carbon emission intensity changes using the Logarithmic Mean Divisia Index model (LMDI) [15]. While there are studies suggesting the importance of integrating the unique aspects of animal husbandry development in various provinces and examining the factors and methods of carbon emissions from animal husbandry based on local conditions, there is currently a lack of literature on this topic. Specifically, there has been no research conducted on the analysis of carbon emissions from animal husbandry in Heilongjiang Province.

In the field of carbon emissions research, there are various analysis methods available. These include the use of Autoregressive Integrated Moving Average model (ARIMA) and neural network prediction models for forecasting carbon emissions [16–19], examining panel data onto different regions in China using spatial econometric models [20,21], studying the decoupling model between economic growth and carbon emissions, and identifying the driving factors that affect carbon emissions [22–26]. When considering carbon sequestration and analyzing issues related to carbon emissions, it is beneficial to combine multiple methods. This approach allows for a more comprehensive understanding of the issue and can lead to more effective solutions. By utilizing various techniques, such as carbon capture and storage, afforestation, and soil carbon sequestration, we can work towards reducing carbon emissions and mitigating the effects of climate change [27–30].

The objective of this study is to integrate the unique features and current states of Heilongjiang Province and adopt the Life Cycle Assessment (LCA) approach to establish a carbon emission evaluation framework of animal husbandry in the region. This framework aims to provide a comprehensive and systematic estimation of carbon emissions associated with animal husbandry activities in Heilongjiang Province. As there is limited research on the low-carbon development of animal husbandry, this study seeks to contribute to the understanding of this topic. Based on estimated results, combined with Kaya's identity and LMDI driver decomposition model, the driving factors affecting carbon emissions from animal husbandry in Heilongjiang Province were discovered. The importance of these factors is analyzed, and problems in the carbon emission process of animal husbandry in Heilongjiang Province are identified. Low carbon measures for animal husbandry are improved to reduce carbon emissions from animal husbandry under the premise of promoting the sustainable development of animal husbandry in Heilongjiang Province. At last, based on the future carbon peak and carbon neutrality trend of animal husbandry in Heilongjiang Province, effective countermeasures and suggestions are proposed to make up for the deficiencies of existing research.

## 2. Materials and Methods

### 2.1. The Calculation Method of Cardon Emission in Animal Husbandry

This research employs the life cycle approach to assess the carbon emissions generated by animal husbandry in Heilongjiang Province. The study focuses on three stages: front-end planting, mid-range breeding, and back-end processing. These stages are further divided into six categories, namely, feed grain planting, feed grain transportation, gastrointestinal

fermentation, fecal management, feeding energy consumption, and livestock product processing (Figure 1). To determine the total carbon emissions from animal husbandry in Heilongjiang Province, we calculated the emissions from each individual link and summed them up. The emissions of greenhouse gas $CH_4$ and $N_2O$ were then converted into carbon dioxide equivalent ($CO_2$-ep) using their global warming potential values.

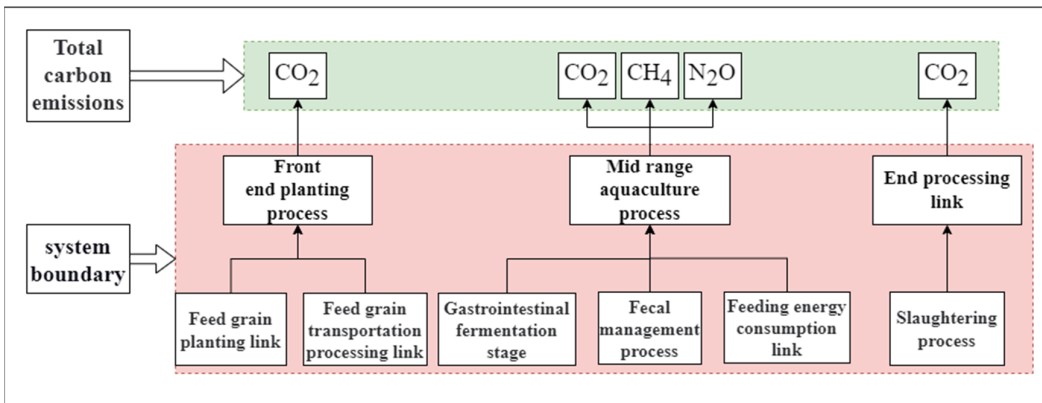

**Figure 1.** Main sources and system boundaries of carbon emissions from animal husbandry in Heilongjiang Province.

### 2.1.1. Front End Planting Process

The carbon emissions from the front-end planting process of livestock and poultry feed include both the feed grain planting process and the associated transportation and processing. These feeds can be classified into two main categories: coarse feed and concentrate feed. The study focuses on the carbon emissions caused by planting concentrate feed, which primarily consists of corn, soybean meal, and wheat bran. The coarse feed, including hay, straw, green feed, and silage, is not included in the calculations due to limited data availability. The formula used for calculating carbon emissions is as follows:

$$E_{CP} = \sum_{u=1}^{n} Q \cdot m \cdot r_u \cdot ef_{u1} . \tag{1}$$

In the formula, $E_{CP}$ represents the carbon dioxide ($CO_2$) emissions generated during the feed grain planting process; Q represents the annual production of livestock products in tons; M represents the grain consumption coefficient per unit product, which refers to the ratio of the grain consumption per animal to the main product yield in kg/kg; $R_u$ represents the proportion of U-type foods in the feed formula; $ef_{u1}$ represents the $CO_2$ emission coefficient during the planting process of U-type grains.

After the raw materials for feed have grown and matured, they undergo processing and are transported to the breeding farm where they are ultimately used to feed livestock and poultry. The calculation formula for this process is as follows:

$$E_{GP} = \sum_{u=1}^{n} Q \cdot m \cdot r_u \cdot ef_{u2} \cdot i_u . \tag{2}$$

In the formula, $E_{GP}$ represents the $CO_2$ emissions generated during the transportation and processing of feed grains; Q represents the annual production of livestock products in tons; M represents the grain consumption coefficient per unit product, which refers to the ratio of the grain consumption per animal to the main product yield in kg/kg; $R_u$ represents the proportion of U-type foods in the feed formula; $ef_{u2}$ represents the $CO_2$ emission coefficient generated during the transportation and processing of U-type feed grains; $I_u$ represents the distribution coefficient of U-type grains.

### 2.1.2. Mid Range Aquaculture Process

The mid range aquaculture process of this study includes the gastrointestinal fermentation process, fecal management process, and feeding energy consumption process.

The calculation formula for the $CH_4$ emissions generated during the gastrointestinal fermentation process of livestock and poultry is

$$E_{ef} = \text{AAP} \cdot ef_1 .$$ (3)

In the formula, $E_{ef}$ represents the $CO_2$ emissions generated during the gastrointestinal fermentation process; $ef_1$ represents the $CH_4$ emission coefficient from gastrointestinal fermentation; AAP represents the average annual feed volume of livestock and poultry in head. When the production cycle of livestock and poultry is greater than or equal to one year (365 days), the year-end inventory quantity of such livestock and poultry is taken as the average annual feed volume. When the production cycle is less than one year (365 days), the annual output data is used to calculate the average annual feed volume. Formula (4) is as follows:

$$APP = \{ \begin{array}{l} Herds_{end}, \, if : Days_{live} \geq 365 \\ Days_{live} \cdot \left( \frac{NAPA}{365} \right), if : Days_{live} < 365 \end{array} .$$ (4)

In the formula, APP refers to the average annual feed volume of livestock and poultry; $Herds_{end}$ is the year-end inventory, head/piece; NAPA is the annual livestock and poultry output, head/animal; $Dat_{live}$ is the average feeding cycle of livestock and poultry in days.

Livestock and poultry manure can produce two types of gases, $CH_4$ and $N_2O$, depending on the conditions. As a result, when calculating carbon emissions form the management of livestock and poultry manure, it is important to consider both of these factors. The following formulas can be used:

$$E_{mc} = \text{AAP} \cdot ef_2 ,$$ (5)

$$E_{md} = \text{AAP} \cdot ef_3 .$$ (6)

In the formula, $E_{mc}$ and $E_{md}$, respectively represent the $CH_4$ and $N_2O$ emissions generated by the fecal management system; AAP represents the average annual livestock and poultry production, unit: head; $ef_2$ and $ef_3$ represent the $CH_4$ and $N_2O$ emission coefficients of the fecal management system, respectively.

Livestock and poultry breeding require specific temperature and humidity conditions, which often results in the consumption of energy sources such as electricity and coal. This, unfortunately, generates $CO_2$ as a byproduct. The formula for this process is as follows:

$$E_{DH} = APP \cdot \frac{cost_\alpha}{price_\alpha} \cdot ef_\alpha + AAP \cdot \frac{cost_\beta}{price_\beta} \cdot ef_\beta .$$ (7)

In the formula: $E_{DH}$ represents the $CO_2$ emissions generated during the energy consumption process of feeding; AAP represents the average annual feeding amount, in head; $cost_\alpha$ and $cost_\beta$, respectively represent the electricity and coal costs consumed by each animal in CNY per animal; $price_\alpha$ and $price_\beta$ represent the unit prices of electricity and coal, respectively. Based on the 2008 Notice on Increasing Electricity Prices in various regions of China, the average agricultural electricity prices in each province are estimated at 0.4275 CNY/(kW·h). However, there is no unified price for heating coal used in aquaculture farms, and it is estimated to be around 800 CNY per ton due to the fact that it is mainly used for heating purposes; $ef_\alpha$ and $ef_\beta$ represent the electricity consumption of each animal and the $CO_2$ emission coefficient generated by coal combustion.

### 2.1.3. End Processing Link

The end processing stage only includes the slaughter processing stage. Livestock and poultry can only be sold in the market after being slaughtered and processed. The calculation formula is as follows:

$$E_{MP} = Q \cdot \frac{MJ}{e} \cdot ef_c .$$

(8)

In the formula, EMP represents the $CO_2$ emissions generated in the livestock product processing process; Q represents the output of livestock products in tons; MJ represents the energy consumption coefficient per unit of livestock product processing; E represents the heat energy generated by consuming one degree of electricity; $ef_C$ represents the $CO_2$ emission coefficient generated by electricity consumption.

### 2.1.4. Summary of Carbon Emissions from Animal Husbandry

According to the full lifecycle assessment method, the formula for calculating the total carbon emissions of animal husbandry in Heilongjiang Province can be expressed as

$$\begin{aligned} E_{TOTAL} = & E_{CP} + E_{GP} + E_{EF} + E_{MM} + E_{DH} + E_{MP} = E_{CP} + E_{GP} + E_{ef} \cdot GWP_{CH_4} + \\ & \left( E_{GP} \cdot GWP_{CH_4} + E_{GP} \cdot GWP_{N_2O} \right) + E_{GP} + E_{GP} . \end{aligned}$$

(9)

In the formula, $E_{TOTAL}$ is the total carbon emissions from animal husbandry in Heilongjiang Province in units of 10 kt/$CO_2$ equivalent; $E_{CP}$, $E_{GP}$, $E_{EF}$, $E_{MM}$, $E_{DH}$, and $E_{MP}$, respectively refer to the carbon emissions generated in the feed grain planting, feed grain transportation and processing, gastrointestinal fermentation, fecal management system, feeding energy consumption, and slaughter processing processes in equivalent units of 10 $ktCO_2$; $GWP_{CH4}$ is the global warming potential of $CH_4$; $GWP_{N2O}$ is the global warming potential value of $N_2O$.

### 2.2. Decoupling Analysis

The term 'decoupling' was first used in physics to describe the gradual detachment of two closely related variables due to certain changes. In the realm of environment and economics, the OECD introduced the concept to examine the evolving relationship between energy consumption and economic growth [31,32]. Decoupling models have been utilized in research across various resource and environmental fields for many years. This article proposes the use of the Tapio decoupling elastic model for decoupling analysis. This model, developed by Professor Petri Tapio, subdivides the decoupling state based on a decoupling index of 1 or −1. The Tapio model offers several advantages over other models including more detailed division and stable results that are not affected by dimensions. The formula for the Tapio decoupling elastic model is as follows:

$$D_{C,G} = \frac{\Delta C / C}{\Delta G / G} .$$

(10)

Among them, $D_{C, G}$ are the decoupling indices between carbon emissions and GDP growth; C is the carbon emissions for the base year; The gross domestic product of animal husbandry in Heilongjiang Province in the base year G; $\Delta C$ is the difference in carbon emissions between the current year and the base year; $\Delta G$ is the difference in GDP between the current year and the base year. The criteria for determining the decoupling elastic model are shown in Table 1.

**Table 1.** Tapio decoupling elastic model judgment criteria.

| Decoupling State | $\Delta CO_2$ | $\Delta GDP$ | Decoupling Index |
|---|---|---|---|
| Expansion negative decoupling (ENDP) | >0 | >0 | E > 1.2 |
| Weak negative decoupling (WND) | <0 | <0 | 0 < E < 0.8 |
| Strong negative decoupling (SND) | <0 | >0 | E < 0 |
| Recessive decoupling (RD) | <0 | <0 | E > 1.2 |
| Weak decoupling (WD) | >0 | >0 | 0 < E < 0.8 |
| Strong decoupling (SD) | >0 | <0 | E < 0 |
| Decay type coupling (STC) | <0 | <0 | 0.8 < E < 1.2 |
| Expansive coupling (EC) | >0 | >0 | 0.8 < E < 1.2 |

### 2.3. Logarithmic Mean Divisia Index

#### 2.3.1. Kaya Identity

The Kaya identity was proposed by Japanese professor Yoichi Kaya during a seminar of the Intergovernmental Panel on Climate Change (IPCC) in 1989. This identity builds upon the IPAT model, which suggests that environmental pressure is the result of the interplay between population, wealth, and technology. Kaya's identity breaks down $CO_2$ emissions into four factors that are tied to human life, development, and production through a simple mathematical formula [33]. The basic formula is as follows:

$$CO_2 = \frac{CO_2}{PE} \times \frac{PE}{GDP} \times \frac{GDP}{P} \times P \, . \tag{11}$$

In the formula, $CO_2$ represents the study $CO_2$ emissions; PE is the energy consumption that causes carbon emissions; GDP is the Gross Domestic Product; P is the total domestic population. After decomposition, $CO_2/PE$ represents $CO_2$. Emission intensity F; PE/GDP represents energy intensity E per unit of GDP; GDP/P represents per capita gross domestic product G; P represents population size, which can be rewritten as a formula after decomposition:

$$CO_2 = F \times E \times G \times P \, . \tag{12}$$

#### 2.3.2. Logarithmic Mean Divisia Index (Region)

In order to assess the impact of carbon emissions on animal husbandry in Heilongjiang Province, the Kaya identity equation has been employed to modify the driving effect. This equation allows for a more precise analysis of the industry's overall results. The expression for this modification is shown in the following formula:

$$H_c = \frac{A_{CE}}{B_{AG}} \times \frac{B_{AG}}{C_{TG}} \times \frac{C_{TG}}{D_{AP}} \times \frac{D_{AP}}{E_{TP}} \times E_{TP} \, . \tag{13}$$

In this formula, Hc and $A_{CE}$ represent the carbon emissions of animal husbandry, while $B_{AG}$ represents the total output value of animal husbandry. $C_{TG}$ represents the total output value of agriculture, forestry, animal husbandry, and fishing. $D_{AP}$ represents the total number of people engaged in agriculture, forestry, animal husbandry, and fishing, and $E_{TP}$ represents the total number of people in Heilongjiang Province. This paragraph discusses the five factors that drive animal husbandry in Heilongjiang Province. These factors are $A_1$, which refers to changes in animal husbandry production efficiency ($A_{CE}/B_{AG}$); $A_2$, which relates to changes in the production structure of animal husbandry ($B_{AG}/C_{TG}$); $A_3$, which concerns changes in agricultural population benefits ($C_{TG}/D_{AP}$); $A_4$, which is linked to the level of population urbanization development ($D_{AP}/E_{TP}$); $A_5$, which refers to the total population of Heilongjiang ($E_{TP}$).

#### 2.3.3. Logarithmic Mean Divisia Index (Livestock)

In order to better reflect the impact of livestock and poultry on carbon emissions in Heilongjiang Province, a modified Kaya identity model has been developed. This model

takes into account the specific driving factors of livestock and poultry in the region, resulting in a more accurate representation of carbon emissions. The resulting decomposition model is as follows:

$$C_y = \frac{C_y}{G_y} \times \frac{G_y}{H_y} \times \frac{H_y}{P_y} \times P_y \,. \tag{14}$$

The equation presented in this study represents the relationship between various factors in the livestock and poultry industry in Heilongjiang Province during a specific period. The variable $C_y$ denotes the total carbon emissions of livestock and poultry in that period, measured in 10,000 tons. $G_y$ represents the output value of the livestock and poultry breeding industry during the same period, measured in billions of CNY. The variable $H_y$ denotes the total output value of animal husbandry in Heilongjiang Province during the period, measured in 100 million CNY. Lastly, the variable $H_y$ represents the number of individuals working in the livestock labor force during the period, measured in 10,000 people. This study aims to decompose the driving factors that affect carbon emissions from animal husbandry in Heilongjiang Province into four types. These include the efficiency factor of animal husbandry ($B_1$), which is measured by the ratio of carbon emissions to animal product output ($C_y/G_y$); the structural factor of the livestock and poultry breeding industry ($B_2$), which is measured by the ratio of animal product output to the number of livestock and poultry ($G_y/H_y$); the economic factors of animal husbandry ($B_3$), which is measured by the ratio of the number of livestock and poultry to the total economic output of animal husbandry ($H_y/P_y$); the labor factor ($B_4$).

### 2.4. Data Sources

The data needed for our research institute have been sourced from reliable publications such as the 'Heilongjiang Province Statistical Yearbook 2001–2021', 'China Rural Statistical Yearbook', and 'National Compilation of Agricultural Product Cost Benefit Information'. Due to the absence of an official carbon emission conversion coefficient for animal husbandry in Heilongjiang Province, we have relied on the coefficients published by the IPCC for estimation purposes. The greenhouse gas emission factor data for each link can be found in Tables 2 and 3. All statistical calculations and result summaries were conducted using Excel software in Microsoft Office 365.

**Table 2.** Greenhouse gas emission factors of livestock and poultry gastrointestinal fermentation and manure management system.

| Livestock and Poultry Breeds | CH$_4$ [kg/(Head. a)] | | N$_2$O [kg/(Head. a)] |
|---|---|---|---|
| | Gastrointestinal Fermentation | Fecal Management | Fecal Management |
| Pig | 1.00 | 3.50 | 0.53 |
| Cattle | 47.80 | 1.00 | 1.39 |
| Cow | 68.00 | 16.00 | 1.00 |
| Buffalo | 55.00 | 2.00 | 1.34 |
| Horse | 18.00 | 1.64 | 1.39 |
| Donkey | 10.00 | 0.90 | 1.39 |
| Mule | 10.00 | 0.90 | 1.39 |
| Sheep | 5.00 | 0.16 | 0.33 |
| Poultry | 0.00 | 0.02 | 0.02 |

**Table 3.** Greenhouse gas emission coefficient of each link of animal husbandry.

| Link | Symbol | Emission Factor | Numerical Value | Unit |
|---|---|---|---|---|
| Planting of feed grains | efu$_1$ | corn | 1.50 | t/t |
| Feed grain transportation and processing | efu$_2$ | corn | 0.0102 | t/t |

**Table 3.** *Cont.*

| Link | Symbol | Emission Factor | Numerical Value | Unit |
|---|---|---|---|---|
| Feed grain transportation and processing | $efu_2$ | soybean | 0.1013 | t/t |
| | | wheat | 0.0319 | t/t |
| Feeding energy consumption | $Ef_1$ | electric energy | 0.9734 | t/(MW·h) |
| | $Price_1$ | electric charge | 0.4275 | CNY/(KW·h) |
| | $Ef_2$ | coal | 1.98 | t/t |
| | $Price_2$ | Unit price of coal | 800 | CNY/t |
| Product processing | $MJ_1$ | pork | 3.76 | MJ/kg |
| | $MJ_2$ | Beef | 4.37 | MJ/kg |
| | $MJ_3$ | Mutton | 10.4 | MJ/kg |
| | $MJ_4$ | Poultry | 2.59 | MJ/kg |
| | $MJ_5$ | Milk | 1.12 | MJ/kg |
| | $MJ_6$ | Poultry eggs | 8.16 | MJ/kg |
| | e | Electric heating value | 3.60 | MJ/(KW·h) |

## 3. Results

### 3.1. Changes in Carbon Emissions from Animal Husbandry

#### 3.1.1. Changes in Total Carbon Emissions from Animal Husbandry

Figure 2 displays the estimated carbon emissions resulting from animal husbandry in Heilongjiang Province from 2000 to 2020. The figure depicts a consistent upward trend, with an average annual growth rate (AAGR%) of approximately 2.289%. The data highlights that the carbon emissions from animal husbandry in Heilongjiang Province can be divided into three distinct stages:

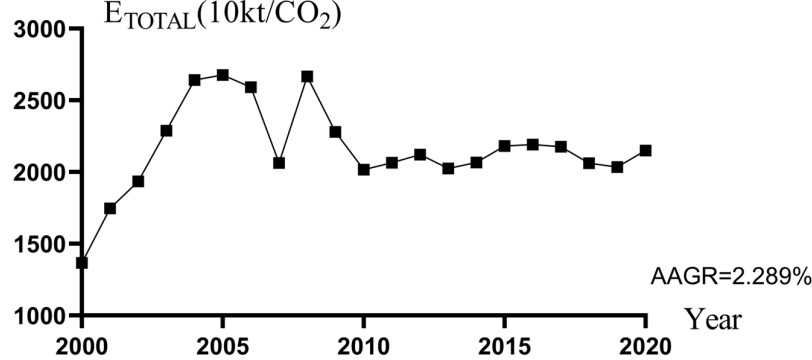

**Figure 2.** The summary unit of carbon emissions from animal husbandry in Heilongjiang Province from 2000 to 2020 is 10 kt/$CO_2$-ep equivalent.

The first stage is a period of rapid development (2000–2008), with carbon emissions ranging from 1366.23 10 kt/$CO_2$ in 2000 to 2665.68 10 kt/$CO_2$ in 2008. During this stage, the animal husbandry industry in Heilongjiang Province has developed rapidly. The government's support through superior subsidy policies has led to a continuous increase in the number of livestock and poultry farmers as well as livestock product processing enterprises in the province. This has resulted in a significant boost in the production of various types of livestock and poultry, as well as livestock products.

The second stage is a relatively stable period (2009–2017), with carbon emissions increasing from 2279.36 10 kt/$CO_2$ in 2009 to 2176.18 10 kt/$CO_2$ in 2017. Compared with the previous stage, the growth rate of carbon emissions from animal husbandry in Heilongjiang Province is relatively stable. The increased government requirements for animal husbandry enterprises and product quality may have led to some enterprises reaching bottlenecks and undergoing a rectification period. This process facilitates the transition towards high-quality animal husbandry.

The third stage is a small decline stage (2018–2020). The carbon emissions show a significant downward trend from 2060.01 10 kt/$CO_2$ in 2018 to 2148.47 10 kt/$CO_2$ in 2020. The national and provincial governments have implemented a comprehensive plan for the layout of Heilongjiang's animal husbandry, with a focus on strengthening supervision, improving relevant policies and laying the foundation for China's 'carbon peak' and 'carbon neutrality' plans. This plan aims to guide livestock and poultry breeding towards a green, healthy, and low-carbon direction, ultimately reducing carbon emissions. However, due to the impact of the novel coronavirus at the end of 2019, carbon emissions in 2020 showed a slight upward trend.

3.1.2. Changes of Carbon Emissions from Animal Husbandry in Different Regions

According to Table 1, it can be seen that in 2000, the carbon emissions of animal husbandry in Harbin, a city in Heilongjiang Province, reached 354.78 10 kt/$CO_2$-ep, ranking first in the province. The Greater Xing'an Mountains region was 5.51 10 kt/$CO_2$-ep, ranking last. In terms of time perspective, all cities except Harbin have a positive average annual growth rate (AAGR) of carbon emissions. Harbin's AAGR is negative at −0.66%. The cities with the highest rising values are Heihe City and Hegang City, with 8.32% and 7.99%, respectively.

The carbon emissions from animal husbandry in Harbin peaked at 721.01 10 kt/$CO_2$-ep in 2015, but have since decreased annually, reaching 310.71 10 kt/$CO_2$-ep in 2019. In contrast, Qiqihar City saw a peak of 536.85 10 kt/$CO_2$-ep in 2016 but has only experienced minor annual decreases in emissions. The carbon emissions from animal husbandry in Jixi City, Qitaihe City, Yichun City, Mudanjiang City, and the Greater Khingan Mountains region were found to be not significantly different in 2019 (54.12 10 kt/$CO_2$-ep, 20.11 10 kt/$CO_2$-ep, 36.41 10 kt/$CO_2$-ep, 102.29 10 kt/$CO_2$-ep, 10.80 10 kt/$CO_2$-ep) compared to 2000 (43.00 10 kt/$CO_2$-ep, 19.92 10 kt/$CO_2$-ep, 23.99 10 kt/$CO_2$-ep, 76.99 10 kt/$CO_2$-ep, and 5.51 10 kt/$CO_2$-ep); In the year 2000, the amount of carbon emissions produced by animal husbandry in four cities were as follows: Hegang City—11.30 10 kt/$CO_2$-ep, Shuangyashan City—16.20 10 kt/$CO_2$-ep, Jiamusi City—52.52 10 kt/$CO_2$-ep, and Heihe City—26.66 10 kt/$CO_2$-ep. From a numerical perspective, it is evident that there has been a significant growth trend with an AAGR of over 4%. The carbon emissions from animal husbandry in Daqing City have remained relatively stable and have exhibited a steady growth trend. Specifically, the emissions have increased from 95.81 10 kt/$CO_2$-ep in 2000 to 251.56 10 kt/$CO_2$-ep in 2019, which is approximately twice the initial amount. The carbon emissions from animal husbandry in Suihua City have been steadily increasing since 2000, starting at 307.28 10 kt/$CO_2$-ep and peaking at 778.23 10 kt/$CO_2$-ep in 2013, the highest value in nearly two decades for various cities in the province. These findings suggest that the animal husbandry industry in this region is growing at a rapid pace and is one of the top livestock producers in the province (Table A1 in Appendix A).

3.1.3. Structural Changes of Carbon Emissions in Animal Husbandry

The data presented in Figure 3 shows that the feed grain planting process ($E_{CP}$) contributes to 30.55% of the total carbon emissions of animal husbandry in Heilongjiang Province, which is roughly one-third of the total. On the other hand, the feed and grain processing and transportation process ($E_{GP}$) only accounts for an average of 0.62% of the total carbon emissions of animal husbandry, with a range of 0.49% to 0.72%. The gastrointestinal fermentation process ($E_{EF}$) is responsible for 42.489% of the total carbon emissions from animal husbandry in the livestock and poultry feeding process. The fecal management process ($E_{MM}$) accounts for 23.15% of the emissions, while the energy consumption process ($E_{DH}$) accounts for 3.16%. On average, $E_{EF}$ accounts for 37.14–49.58% and $E_{MM}$ accounts for 21.48–25.23%. The feeding process accounts for approximately two-thirds (68.81%) of the total, while slaughtering and processing ($E_{MP}$) only account for a small proportion (0.02–0.03%, averaging at 0.03%).

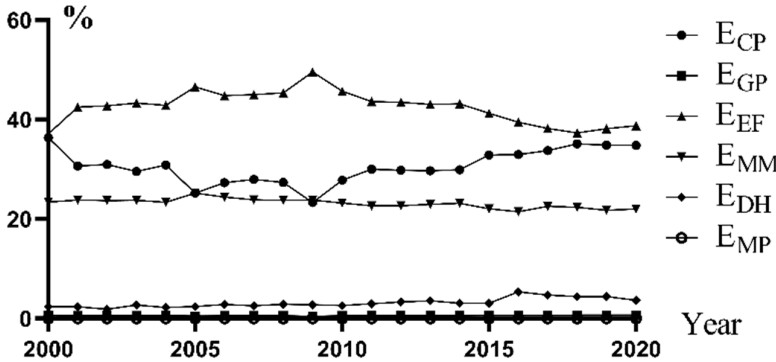

**Figure 3.** Carbon emissions from various links of animal husbandry in Heilongjiang Province.

3.1.4. Changes of Carbon Emissions of Livestock and Poultry

Figure 4 displays the proportion of carbon emissions from livestock and poultry in Heilongjiang Province for four years. The highest proportion of carbon emissions from livestock and poultry in Heilongjiang Province was observed in beef cattle followed by pigs, cows, poultry, sheep, horses, donkeys, and mules in 2005. According to data from 2010, 2015, and 2020, cows, beef cattle, and pigs were found to have the highest carbon emissions among livestock and poultry. However, the proportion of carbon emissions from horses, donkeys, and mules decreased annually with a set limit.

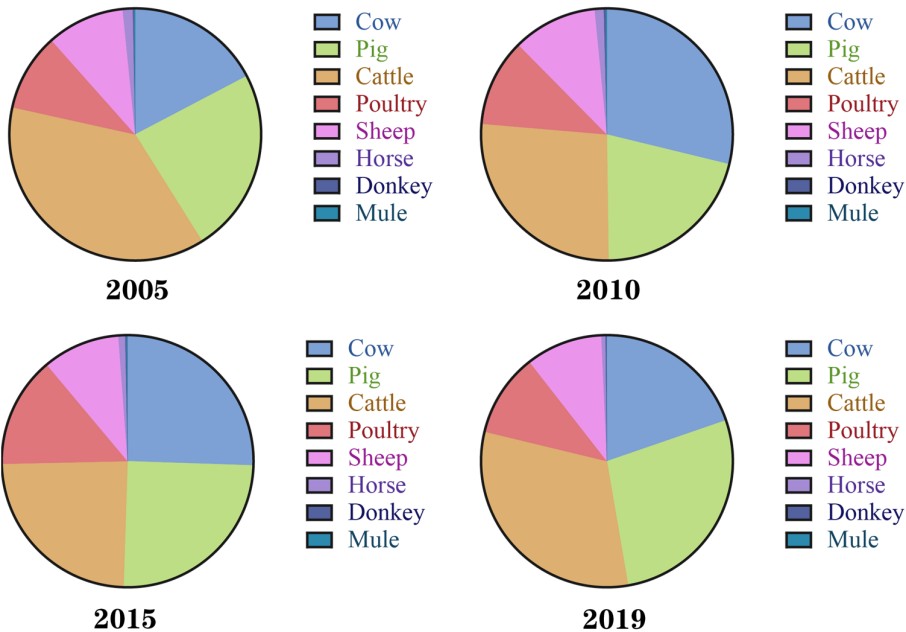

**Figure 4.** Pie chart of the proportion of livestock and poultry carbon emissions in Heilongjiang Province.

### 3.2. *Tapio Decoupling Analysis Results*

Analysis of Decoupling State in Heilongjiang Province

After incorporating the carbon emissions and total output value of animal husbandry in Heilongjiang Province from 2000 to 2020 into the Tapio decoupling elasticity model, we were able to obtain the decoupling elasticity index of Heilongjiang Province for each year from 2000 to 2020. The results of this analysis are presented in Table 4.

**Table 4.** Decoupling elasticity index of Heilongjiang Province from 2000 to 2020.

| Year | $\Delta CO_2$ | $\Delta GDP$ | E | Decoupling Condition [1] |
|---|---|---|---|---|
| 2000–2001 | 0.28 | 0.19 | 1.47 | ENDP |
| 2001–2002 | 0.11 | 0.10 | 1.13 | EC |
| 2002–2003 | 0.18 | 0.13 | 1.43 | ENDP |
| 2003–2004 | 0.16 | 0.27 | 0.57 | WD |
| 2004–2005 | 0.01 | 0.15 | 0.09 | WD |
| 2005–2006 | −0.03 | −0.03 | 1.17 | STC |
| 2006–2007 | −0.20 | 0.27 | −0.77 | SND |
| 2007–2008 | 0.29 | 0.40 | 0.73 | WD |
| 2008–2009 | −0.15 | 0.06 | −2.43 | SND |
| 2009–2010 | −0.12 | 0.10 | −1.19 | SND |
| 2010–2011 | 0.02 | 0.31 | 0.08 | WD |
| 2011–2012 | 0.03 | 0.08 | 0.37 | WD |
| 2012–2013 | −0.05 | 0.02 | −2.45 | SND |
| 2013–2014 | 0.02 | 0.01 | 9.41 | ENDP |
| 2014–2015 | 0.06 | 0.11 | 0.52 | WD |
| 2015–2016 | 0.01 | 0.06 | 0.08 | WD |
| 2016–2017 | −0.01 | 0.04 | −0.18 | SND |
| 2017–2018 | −0.05 | −0.10 | 0.52 | STC |
| 2018–2019 | −0.01 | 0.09 | −0.14 | SND |
| 2019–2020 | 0.06 | 0.14 | 0.39 | WD |

[1] Expansion negative decoupling (ENDP); Weak negative decoupling (WND); Strong negative decoupling (SND); Recessive decoupling (RD); Weak decoupling (WD); Strong decoupling (SD); Decay type coupling (STC); Expansive coupling (EC).

The relationship between carbon emissions from animal husbandry and the development of the animal husbandry economy in Heilongjiang Province can be divided into four stages. The first stage, from 2000 to 2003, focused on expanding negative decoupling. The development of animal husbandry in Heilongjiang Province has led to a significant increase in carbon emissions, highlighting the high demands placed on the animal husbandry economy in the region. The second stage, spanning from 2003 to 2013, was primarily marked by weak decoupling and strong negative decoupling. During this period, there was a concerted effort to shift the animal husbandry industry in Heilongjiang Province towards energy conservation and emission reduction. Weak decoupling refers to the phenomenon where carbon emissions increase at a slower rate than economic growth, whereas strong negative decoupling occurs when carbon emissions rise during an economic downturn. During the period from 2013 to 2017, the situation of decoupling was relatively complex, with weak decoupling and strong negative decoupling being the main factors. However, it was observed that the weak decoupling effect was significant from 2014 to 2016 with a low elasticity index, indicating that the low-carbon development of animal husbandry in Heilongjiang Province had achieved certain results. Although a strong negative decoupling was observed between 2016 and 2017, the decoupling elasticity index was found to be insignificant. Additionally, previous research has shown that the number of large livestock raised sharply decreased during this time period, which is believed to be the main reason for the observed strong negative decoupling. Between 2017 and 2020, there was a state of recession coupled with decoupling, as the number of livestock and poultry farming decreased from 2017 to 2018. This state suggests that carbon emissions decreased at a moderate rate during the economic recession, while the overall number of livestock and poultry farming began to increase from 2018 to 2019 and reached a high level in 2020. The decoupling state in Heilongjiang Province exhibited weak decoupling from 2019 to 2020. This suggests that the province has made significant progress in energy conservation and emission reduction in the animal husbandry industry. Currently, the animal husbandry sector in Heilongjiang is at a crucial stage of transition to strong decoupling.

### 3.3. Analysis of Influencing Factors of Carbon Emission in Animal Husbandry

3.3.1. Analysis on Influencing Factors of Animal Husbandry in Heilongjiang Province

The LMDI driving factor models indicate that the change in livestock production efficiency ($\Delta A_1$) is the primary driving factor for reducing carbon emissions in the Heilongjiang Province's livestock industry. The average change in carbon emissions attributed to this factor is 2.1938 million tons, as shown in Figure 5. After comparing the production efficiency factors in Heilongjiang Province from 2000 to 2020, it is evident that the province, known for its agricultural and pastoral areas and being a major grain producer in China, has been actively promoting the concentration of animal husbandry in favorable regions. This has led to the continuous growth of animal husbandry in the province, moving towards a more scaled and intensified approach. To reduce carbon emissions and improve the output value of animal husbandry in Heilongjiang Province, it is recommended to establish large and medium-sized animal waste treatment plants in suitable locations. This highlights the importance of animal husbandry production efficiency in mitigating carbon emissions, particularly in major animal husbandry provinces. The level of population urbanization development ($\Delta A_4$) and the total population of Heilongjiang Province ($\Delta A_5$) are two additional factors that also contribute to the reduction of carbon emissions from animal husbandry. Specifically, these factors are estimated to result in a reduction of 0.239 million tons and 0.192 million tons, respectively. The level of urbanization in Heilongjiang Province has been steadily reducing carbon emissions from animal husbandry since 2011, suggesting a decline in the number of individuals involved in agriculture, forestry, animal husbandry, and fishing each year. The reduction of employees in animal husbandry is primarily affecting individual farmers. This reduction can lead to the promotion of local animal husbandry's intensive development, resulting in a decrease in carbon emissions. However, the total population loss in Heilongjiang Province each year can create a negative driving force for carbon emissions in animal husbandry. Agriculture continues to be a significant contributor to the primary, secondary, and tertiary sectors in Heilongjiang Province. As a result, a portion of the decreasing population relies heavily on agricultural development for their livelihood. Additionally, the reduction in population also leads to a decrease in carbon emissions.

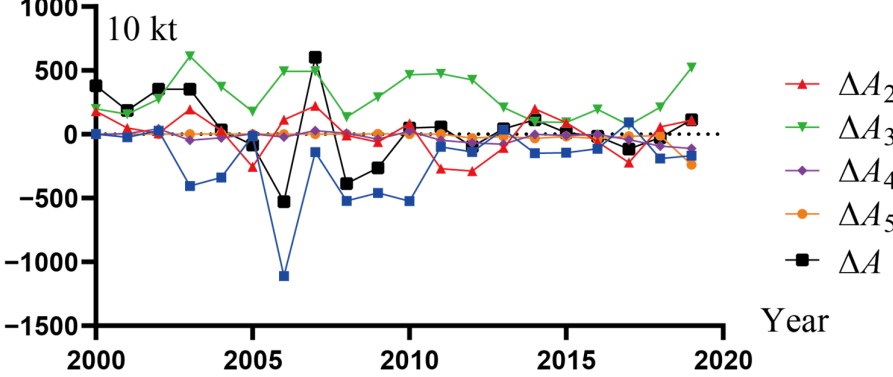

**Figure 5.** Analysis results of driving factors of carbon emissions from animal husbandry in Heilongjiang Province (2000–2020).

The main factor contributing to carbon emissions from animal husbandry is the increase in agricultural population income ($\Delta A_3$), which has an average annual growth rate of 2.9834 million tons. This has been identified as the main driving force for the continuous rise in carbon emissions from animal husbandry in Heilongjiang Province. The higher production income of animal husbandry compared to planting has led farmers in Heilongjiang Province to expand their animal husbandry production, resulting in increased annual income. However, this expansion has also caused a significant rise in carbon emissions from animal husbandry. While the increase in farmers' income is a positive outcome, it is important to consider the impact on carbon emissions in animal husbandry

in Heilongjiang Province. Achieving carbon neutrality requires a careful balance between increasing farmers' income and reducing carbon emissions. Therefore, it is crucial to find ways to effectively manage this relationship in order to promote sustainable development in the region. The production structure of animal husbandry ($\Delta A_2$) has undergone a significant change, resulting in an increase in production by 0.032 million tons. While there was mainly inhibition in 2018, the impact of the novel coronavirus in 2019 has led to a serious impact on the tourism industry and related services, as well as other tertiary industries. This has resulted in an increase in carbon emissions from animal husbandry.

3.3.2. Analysis of Main Influencing Factors of Livestock and Poultry

This study focuses on the driving factors behind the carbon emissions of beef cattle, cows, and pigs in Heilongjiang Province, based on previous research that identified these as the livestock with the highest emissions. Other types of livestock are not included in the analysis. The research findings suggest that economic factors are the primary driver of carbon emissions from livestock and poultry in Heilongjiang Province. Specifically, the increase in livestock income is positively correlated with the rise in carbon emissions from animal husbandry. Efficiency, structural, and labor factors are all contributing to the limitation of carbon emissions in livestock and poultry. However, it has been observed that cows have lower efficiency factors compared to live pigs and beef cattle. This is due to the fact that cows are milked 3–4 times a day during the production process, resulting in varying levels of efficiency across farms and cows. Cows produce larger carbon emissions compared to live pigs and beef cattle. Efficiency factors do not have a significant impact on reducing carbon emissions during cow production, while structural factors have a greater influence on cows than on live pigs and beef cattle. The dairy farming industry is undergoing structural changes that will have a significant impact on carbon emissions. As a result, the most effective way to reduce carbon emissions from cows in Heilongjiang Province is to adjust the industrial structure and work with appropriate labor and technology. The driving factors for live pigs and beef cattle show similar results, as shown in Table 5.

**Table 5.** Results of main livestock and poultry driving factors of livestock carbon emissions in Heilongjiang Province.

| Year | $\Delta B_1$ [1] | $\Delta B_2$ | $\Delta B_3$ | $\Delta B_4$ | $\Delta B$ |
|------|------|------|------|------|------|
| Cow | −20.49 | −25.54 | 134.12 | −79.78 | 44.33 |
| Pig | −50.98 | −1.39 | 140.16 | −80.48 | 7.32 |
| Cattle | −58.78 | −17.46 | 156.16 | −78.07 | 1.86 |

[1] $\Delta B_1$ is the efficiency factor of animal husbandry; $\Delta B_2$ is the structural factor of livestock and poultry; $\Delta B_3$ is the economic benefit factor for livestock and poultry; $\Delta B_4$ is labor force factor; $\Delta B$ represents the total effect.

## 4. Conclusions and Policy Implications

### 4.1. Conclusions

This article aims to estimate the carbon emissions of animal husbandry in Heilongjiang Province and 13 cities from 2000 to 2020 using the Life Cycle Assessment (LCA) method. The study utilizes the Tapio decoupling model, Kaya identity, and LMDI model to explore the relationship between the animal husbandry economy and carbon emissions in Heilongjiang Province, as well as the driving factors affecting animal husbandry carbon emissions. The results of the study provide valuable insights into the carbon emissions of animal husbandry in the region.

(1) The carbon emissions of animal husbandry in Heilongjiang Province have been on an overall upward trend compared to 2000, with an average annual growth rate of 2.289%. According to LCA estimation results, the gastrointestinal fermentation stages (43.10%), feed grain planting stage (30.58%), and fecal management stage (22.48%) are the stages where the most carbon emissions are generated. This indicates significant differences in carbon emissions generated at different feeding stages. This is consistent with the

study by Sudarshan Mahala et al. [34]. When searching for effective ways to decrease carbon emissions in animal husbandry in Heilongjiang Province, it is recommended to prioritize the stage with the highest levels of carbon emissions. According to the panel data on carbon emissions in animal husbandry in Heilongjiang Province from 2000 to 2019, it has been observed that the carbon emissions of animal husbandry in Daqing City have been steadily increasing over the years. This trend suggests that animal husbandry in Heilongjiang Province is gradually shifting towards other regions such as Suihua, Qiqihar, and Daqing City, thereby establishing itself as a favorable area for the development of animal husbandry in the province.

(2)  The decoupling results in Heilongjiang Province are positive, demonstrating a stable trend of weak decoupling, steady economic growth, and a gradual increase in carbon emissions.

(3)  In the context of Heilongjiang Province and major carbon-emitting regions, the primary factors that contribute to the reduction of carbon emissions from animal husbandry are improvements in production efficiency, the level of urbanization development, and population growth. The study identifies changes in agricultural population returns and the production structure of animal husbandry as the driving factors affecting carbon emissions. Efficiency, structural, and labor factors are the main drivers for reducing carbon emissions in cows, pigs, and beef cattle, while economic factors promote carbon emissions.

### 4.2. Policy Implications

(1)  In terms of carbon emissions, gastrointestinal fermentation (43.10%), feed grain cultivation (30.58%), and fecal management (22.48%) are the three major contributors. As a result, it is recommended that the government increase support for initiatives that enhance soil carbon sequestration capacity, improve animal husbandry technology, and promote genetic breeding advancements. Harindintwali JD's [35] research results suggest that the production of nitrogen can be reduced by optimizing traditional organic fertilizer compost to replace chemical nitrogen fertilizer. This indicates that by doing so, we can move towards more sustainable and ecofriendly animal husbandry.

(2)  While the energy consumption of feed only contributes to 3.16% of total carbon emissions, regulating and reducing emissions in this area is relatively straightforward. To improve the overall literacy of staff working in livestock and poultry farms, it is recommended to provide centralized training sessions that focus on guiding energy conservation and promote green and low-carbon ideas. This will help to prevent indiscriminate waste of energy, such as electricity, in the feeding environment. Additionally, it is important to establish a low-carbon concept for livestock farms.

(3)  In light of the persistent increase in carbon emissions in Heilongjiang Province, research has revealed that animal husbandry in the province is predominantly concentrated in the areas of Daqing, Qiqihar, and Suihua. This concentration highlights the regions where animal husbandry is most advantageous in Heilongjiang Province. In order to ensure the sustainable development of animal husbandry in this region, it is necessary to promote its transfer within Heilongjiang Province while continuously improving its scale and intensification level. To achieve energy-saving and emission reduction effects, it is recommended to build animal waste treatment plants in adjacent areas for convenient centralized treatment of livestock and poultry waste.

(4)  The decoupling status between carbon emissions from animal husbandry and economic growth in Heilongjiang Province is mainly weak decoupling, indicating that while the economy is developing steadily, carbon emissions are slowly increasing. To achieve a 'strong decoupling' of carbon emissions from animal husbandry in Heilongjiang Province, it is recommended to utilize a decoupling model to monitor the carbon emissions from animal husbandry in real-time. This will help control the rapid growth of $CO_2$ in animal husbandry and ensure the decoupling status is maintained.

(5) The increase in carbon emissions from animal husbandry in Heilongjiang Province is primarily driven by changes in agricultural population returns and production structure. On the other hand, factors such as production efficiency, population urbanization development level, and population mobility have a suppressing effect on carbon emissions. Livestock and poultry farming may be the key to poverty alleviation for some private farmers. It is important to consider the methods and scales of breeding and avoid forcing farmers to engage in intensive and concentrated breeding. This can be achieved by analyzing the driving decomposition results. To reduce income disparities in the livestock industry across regions, it is recommended to standardize management practices. Additionally, it is important to tailor methods to individual private farmers, taking into account the location and quantity of their livestock and poultry. This can be achieved by adjusting production structures in a reasonable manner and implementing waste treatment facilities centrally with uniform management. To improve breeding efficiency, it is recommended to increase the frequency of professional training and enhance the level of high-quality livestock and poultry breeding among farmers.

(6) Livestock farms are typically located in remote areas with flat terrain, providing an opportunity to utilize the surrounding idle land. While there are numerous methods for fixing $CO_2$, tree planting's carbon fixation cycle may be longer, but it offers advantages such as ease of operation and low cost, making it a sustainable option. Based on the geographical location and climate environment of Heilongjiang Province, it is advisable to carefully choose appropriate plants to be planted in idle land. This not only leads to a reasonable carbon fixation effect but also enhances the living environment of livestock farms. Studies have shown that livestock and poultry can exhibit better production potential in a natural environment [36]. Due to the open geographical location of the livestock farm, there are fewer buildings around it, resulting in longer lighting time. This presents an opportunity to promote the use of photovoltaic power generation technology, which can be implemented within the farm to provide clean energy [37].

(7) In order to promote environmental protection in animal husbandry, it is essential to provide both policy and financial support for waste management projects. Unfortunately, the high prices and usage costs of waste treatment equipment pose a significant challenge for small and medium-sized farms [38]. As a result, it is difficult for these farms to sustain the long-term use of such equipment. Therefore, policymakers should consider implementing measures to reduce the financial burden of implementing environmental protection measures in animal husbandry, such as offering subsidies or tax incentives. To address this issue, the government can take two measures. Firstly, encouraging innovation and improvement in animal husbandry environmental protection projects can lead to the development of low-cost and effective new equipment, specifically in areas such as animal welfare [39], $CO_2$ biological storage [40], and nitrogen emission microbial treatment [41]. Additionally, the government can provide policy subsidies or cost subsidies to evaluate the construction of environmental protection projects in aquaculture farms.

(8) The present article acknowledges certain limitations. Firstly, as the data is solely based on yearbook data, there may be some discrepancies in actual production while estimating carbon emissions from animal husbandry. Secondly, while conducting factor analysis from three dimensions, namely province, region, and livestock and poultry, the specific implementation of policies is often carried out in smaller administrative units or cities. Thus, future research should focus on narrowing down the research area to provide a more precise reference for the sustainable development of animal husbandry and carbon emissions in various counties and cities.

**Author Contributions:** Data curation, T.H.; investigation, X.L.; formal analysis, T.H.; writing—original draft, T.H. and C.W.; writing—review and editing, C.W.; supervision, Y.Q.; validation, T.H.; visualization, T.H. All authors have read and agreed to the published version of the manuscript.

**Funding:** Heilongjiang Province's "Million and Ten Million" Major Project in Science and Technology (2021ZX12B03).

**Institutional Review Board Statement:** Not applicable.

**Informed Consent Statement:** Not applicable.

**Data Availability Statement:** The datasets used and analyzed during the current study are available from the corresponding author on reasonable request.

**Conflicts of Interest:** The authors declare no conflict of interest.

## Appendix A

**Table A1.** Carbon emissions of animal husbandry in Heilongjiang Province from 2000 to 2019 (unit: 10 kt/$CO_2$ equivalent).

| Year | Harbin | Qiqihar | Jixi | Hegang | Shuangyashan | Daqing | Yichun | Jiamusi | Qitaihe | Mudanjiang | Heihe | Suihua | Da Hinggan Ling |
|---|---|---|---|---|---|---|---|---|---|---|---|---|---|
| 2000 | 354.78 | 249.13 | 43.00 | 11.30 | 16.20 | 95.81 | 23.99 | 52.52 | 19.93 | 77.00 | 26.66 | 307.28 | 5.51 |
| 2001 | 456.35 | 287.46 | 60.91 | 18.14 | 25.80 | 111.90 | 30.69 | 97.21 | 28.18 | 107.78 | 34.83 | 343.03 | 8.40 |
| 2002 | 473.91 | 323.72 | 65.86 | 20.47 | 31.75 | 145.69 | 35.47 | 115.15 | 31.39 | 117.83 | 41.98 | 362.32 | 8.27 |
| 2003 | 533.33 | 376.36 | 78.47 | 25.21 | 41.76 | 178.74 | 41.22 | 131.10 | 39.79 | 135.00 | 55.45 | 416.15 | 16.68 |
| 2004 | 597.65 | 442.76 | 84.51 | 27.48 | 63.65 | 206.43 | 49.47 | 146.62 | 49.03 | 141.15 | 62.72 | 495.20 | 22.33 |
| 2005 | 569.30 | 446.31 | 85.22 | 28.52 | 91.28 | 209.62 | 52.65 | 151.47 | 49.33 | 145.50 | 68.54 | 485.82 | 18.31 |
| 2006 | 548.23 | 436.03 | 78.14 | 25.14 | 96.01 | 213.69 | 50.23 | 153.67 | 52.99 | 127.49 | 56.75 | 422.14 | 13.24 |
| 2007 | 401.79 | 330.60 | 42.65 | 13.26 | 78.60 | 182.17 | 42.78 | 117.90 | 16.74 | 89.18 | 53.00 | 380.49 | 8.66 |
| 2008 | 569.71 | 428.96 | 53.90 | 22.31 | 110.48 | 228.05 | 50.50 | 162.25 | 22.85 | 98.68 | 76.96 | 548.69 | 10.84 |
| 2009 | 603.03 | 445.97 | 54.77 | 25.19 | 122.26 | 255.62 | 54.70 | 178.51 | 22.01 | 107.47 | 85.23 | 599.38 | 11.39 |
| 2010 | 632.16 | 482.46 | 57.51 | 26.36 | 135.15 | 286.69 | 56.77 | 194.04 | 24.67 | 113.29 | 98.08 | 700.31 | 11.99 |
| 2011 | 643.03 | 508.73 | 64.55 | 29.72 | 149.73 | 325.39 | 57.99 | 219.84 | 27.11 | 119.37 | 114.60 | 724.39 | 12.47 |
| 2012 | 678.25 | 561.05 | 69.63 | 31.46 | 138.20 | 352.32 | 63.58 | 249.33 | 32.44 | 126.92 | 131.98 | 752.59 | 13.09 |
| 2013 | 703.03 | 572.78 | 74.03 | 28.38 | 139.42 | 356.64 | 65.47 | 257.18 | 30.86 | 137.19 | 142.30 | 778.23 | 13.82 |
| 2014 | 718.22 | 587.13 | 77.12 | 21.45 | 81.87 | 357.37 | 65.62 | 278.93 | 31.09 | 145.95 | 153.38 | 745.64 | 14.64 |
| 2015 | 721.01 | 521.94 | 80.28 | 20.91 | 65.10 | 269.32 | 65.11 | 287.59 | 24.75 | 149.27 | 162.26 | 718.63 | 15.44 |
| 2016 | 684.33 | 536.85 | 74.65 | 19.88 | 57.03 | 257.65 | 59.38 | 298.78 | 25.37 | 150.21 | 161.94 | 707.12 | 16.40 |
| 2017 | 410.42 | 404.98 | 45.48 | 19.48 | 30.64 | 214.22 | 39.24 | 136.71 | 22.69 | 105.10 | 135.45 | 503.53 | 11.55 |
| 2018 | 339.96 | 395.90 | 64.29 | 64.92 | 42.66 | 233.57 | 39.97 | 142.50 | 20.56 | 101.88 | 153.35 | 455.43 | 9.49 |
| 2019 | 310.71 | 388.41 | 54.12 | 52.60 | 38.22 | 251.56 | 36.41 | 145.55 | 20.11 | 102.29 | 131.94 | 422.92 | 10.80 |
| AAGR [1] | −0.66% | 2.25% | 1.16% | 7.99% | 4.39% | 4.95% | 2.11% | 5.23% | 0.05% | 1.43% | 8.32% | 1.61% | 3.42% |

[1] AAGR stands for average annual growth rate.

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
