# Peer review of "Study on the Relationship between Economic Growth of Animal Husbandry and Carbon Emission Based on Logarithmic Average Index Method and Decoupling Model: A Case Study of Heilongjiang Province"

_sustainability, doi:10.3390/su15139964_

Round 1
Reviewer 1 Report
This is a well written manuscript that does a good job of evaluating the relationship between economic growth of animal husbandry and carbon emission. However, I have a few concerns that are listed below.
Equation 7: How were the prices calculated? What data was used?
Results:
Table 4,5, 6 should be converted to figures.
Conclusions: Policy Implications:
The policies listed here need to be compared against existing literature with discussion about policy intersection and divergence. Supporting references need to be appropriately cited. For example for policy implication 7.: Nisal, Apoorva, et al. "Evaluation of global techno-socio-economic policies for the FEW nexus with an optimal control based approach." Frontiers in Sustainability 3 (2022): 948443.
Assumptions and limitations of this study need to be included.
Reviewer 2 Report
Comments:
Write full form of used abbreviation once it appears first like OECD etc:
The uses of statistical tools are lacking in the methodology and results tables. Need to improve for better representation of repeatability of the results;
Write the full form of the abbreviation at the bottom of the table for self-explanatory;
The cited literatures are old, need to cite most recent papers and delete old references;
Minor editing is needed
Reviewer 3 Report
er: sustainability-2391451
Journal: Sustainability
It is my pleasure to review the manuscript for the Sustainability, an uprising journal. In the manuscript of “Study on the relationship between economic growth of animal husbandry and carbon emission based on logarithmic average index method and decoupling model: A case study of Heilongjiang Province”, the authors estimated the carbon emissions of animal husbandry in Heilongjiang Province and cities from 2000 to 2020 using the Life Cycle Assessment (LCA) method, and then, investigated the relationship between animal husbandry economy and carbon emissions in Heilongjiang Province and the driving factors affecting animal husbandry carbon emissions using Tapio decoupling model, Kaya identity, and Log- Divisia Index Method model. The work presented is relevant to the Journal's field. The manuscript has got some potential. I would like to congratulate the author for a considerable amount of work that they have done. Especially, the authors reported that the main driving force behind the continuous increase in carbon emissions from animal husbandry in Heilongjiang Province is the changing factors of agricultural population returns and changes into production structure of animal husbandry; The driving factors that suppress the increase in carbon emissions from animal husbandry in Heilongjiang Province are changes in animal husbandry production efficiency, population and urban development levels, and population mobility factors. This manuscript has provided a new case to a better understanding of the relationship between economic growth of animal husbandry and carbon emission in emerging countries, such as China. However, the manuscript needs further improved before to be accepted for publication. The reviewer has listed some specific comments that might be helpful of the author to further enhance the quality of the manuscript. Please consider the particular comments listed below.
Comment 1: Abstract. The description of the study context/importance of the study is relatively good, the study methods and data are clearly described, and the results are summarised in an acceptable manner. However, it should further underscore the scientific value added of your paper.
Comment 2: section of Introduction. Although the section is well-structured and well-organized, the novelty of this paper should be further justified by highlighting main contributions to the existing introduction. This could be clearly presented in the Literature review related work. Please consider citing following papers entitled “Revisiting the environmental kuznets curve hypothesis in 208 counties: The roles of trade openness, human capital, renewable energy and natural resource rent”; next entitled “Per-capita carbon emissions in 147 countries: The effect of economic, energy, social, and trade structural changes”, next entitled “Does urbanization redefine the environmental Kuznets curve? An empirical analysis of 134 Countries”, and next titled “Does renewable energy reduce ecological footprint at the expense of economic growth? An empirical analysis of 120 countries”, and next titled “The impact of energy efficiency on carbon emissions: Evidence from the transportation sector in Chinese 30 provinces”. There has already been a large number of literatures related to your research topic, i.e., influencing factors of carbon emission. In addition, there are also have been another larger number of literatures related to your research technique/approach/model, i.e., time-series decomposition. Therefore, it should be better elaborate the contribution of the work to the existing literature, so as to further bridge the gaps between the research background and research purposes.
Comment 3: section of materials and methods. The section is well-structured and well-written. However, it would be better to further highlight your improvement of the method and your innovation in methods.
Comment 4: sections of results, Discussion and policy implications. These two sections are also well-structured and well-organized. However, it would be better to discuss what your findings are different from the past works. A comparison with the results of the previous paper would further enhance the innovative nature of the paper
Comment 5: section of conclusion and policy implications. Please make sure your conclusions' section underscore the scientific value added of your paper, and/or the applicability of your findings/results, as indicated previously. Basically, you should enhance your contributions, limitations, underscore the scientific value added of your paper, and/or the applicability of your findings/results and future study in this session.
Comment 6: There are still some occasional grammar errors through the revised manuscript especially the article ''the'', ''a'' and ''an'' is missing in many places, please make a spellchecking in addition to these minor issues. In addition, some sentences are too long to be easy to read. It is recommended to change to short sentences, which are easier to read.
Comment 7: References. Please check the references in the text and the list; You should update the reference. Please read the latest published papers carefully and format your references according to the format required by Sustainability. If this revised paper is sent to me for re-review, the first thing I will check the references.
Moderate editing of English language required
Round 2
Reviewer 3 Report
Accept in present form